# Role of MPK4 in pathogen-associated molecular pattern-triggered alternative splicing in Arabidopsis

Jeremie Bazin[1,2◦], Kiruthiga Mariappan[3◦], Yunhe Jiang[3], Thomas Blein[1,2], Ronny Voelz[3], Martin Crespi[1,2], Heribert Hirt[3,4]*

1 Université Paris Saclay, CNRS, INRAE, Univ Evry, Institute of Plant Sciences Paris Saclay (IPS2), Orsay, France, 2 Université de Paris, CNRS, INRAE, Institute of Plant Sciences Paris Saclay (IPS2), Orsay, France, 3 DARWIN21, Biological and Environmental Sciences and Engineering Division, King AbdullahUniversity of Science and Technology (KAUST), Thuwal, Saudi Arabia, 4 Max F. Perutz Laboratories, University of Vienna, Dr. Bohrgasse, Vienna, Austria

◦ These authors contributed equally to this work.
* heribert.hirt@kaust.edu.sa

**Data Availability Statement:** All data are available at the Short Read Archive database (https://www.ncbi.nlm.nih.gov/sra/) under the following accessions: PRJNA379910, GSE146189.

## Abstract

Alternative splicing (AS) of pre-mRNAs in plants is an important mechanism of gene regulation in environmental stress tolerance but plant signals involved are essentially unknown. Pathogen-associated molecular pattern (PAMP)-triggered immunity (PTI) is mediated by mitogen-activated protein kinases and the majority of PTI defense genes are regulated by MPK3, MPK4 and MPK6. These responses have been mainly analyzed at the transcriptional level, however many splicing factors are direct targets of MAPKs. Here, we studied alternative splicing induced by the PAMP flagellin in Arabidopsis. We identified 506 PAMP-induced differentially alternatively spliced (DAS) genes. Importantly, of the 506 PAMP-induced DAS genes, only 89 overlap with the set of 1950 PAMP-induced differentially expressed genes (DEG), indicating that transcriptome analysis does not identify most DAS events. Global DAS analysis of *mpk3*, *mpk4*, and *mpk6* mutants in the absence of PAMP treatment showed no major splicing changes. However, in contrast to MPK3 and MPK6, MPK4 was found to be a key regulator of PAMP-induced DAS events as the AS of a number of splicing factors and immunity-related protein kinases is affected, such as the calcium-dependent protein kinase CPK28, the cysteine-rich receptor like kinases CRK13 and CRK29 or the FLS2 co-receptor SERK4/BKK1. Although MPK4 is guarded by SUMM2 and consequently, the *mpk4* dwarf and DEG phenotypes are suppressed in *mpk4 summ2* mutants, MPK4-dependent DAS is not suppressed by SUMM2, supporting the notion that PAMP-triggered MPK4 activation mediates regulation of alternative splicing.

## Author summary

Alternative splicing (AS) of pre-mRNAs in plants is an important mechanism of gene regulation in environmental stress tolerance but plant signals involved are essentially unknown. Pathogen-associated molecular pattern (PAMP)-triggered immunity (PTI) is

**Funding:** This publication is based upon work supported by the King Abdullah University of Science and Technology (KAUST) to Prof. Heribert Hirt No. BAS/1/1062-01-01. This work was supported by grants of The King Abdullah University of Science and Technology (KAUST) International Program OCRF-2014-CRG4 to Prof Heribert Hirt and Dr. Martin Crespi. Dr Martin Crespi benefits from the support of Saclay Plant Sciences-SPS (ANR-17-EUR-0007). The funders had no role in study design, data collection and analysis, decision to publish, or preparation of the manuscript.

**Competing interests:** The authors have declared that no competing interests exist.

mediated by mitogen-activated protein kinases and the majority of PTI defense genes are regulated by MPK3, MPK4 and MPK6. These responses have been mainly analyzed at the transcriptional level, however many splicing factors are direct targets of MAPKs. Here, we studied PAMP-induced alternative splicing in Arabidopsis and identified several hundred differentially alternatively spliced (DAS) genes. Importantly, of these PAMP-induced DAS genes, only 18% overlap with the set of PAMP-induced differentially expressed genes (DEG), indicating that transcriptome analysis does not identify most DAS events. Global DAS analysis of MAPK mutants identified MPK4 as a key regulator of PAMP-induced DAS events. Although MPK4 is guarded by SUMM2 and consequently, the *mpk4* dwarf and DEG phenotypes are suppressed in *mpk4 summ2* mutants, MPK4-dependent DAS is not suppressed by SUMM2, showing that PAMP-triggered MPK4 activation mediates regulation of alternative splicing.

## Introduction

Plants possess pattern recognition receptors that detect conserved pathogen-associated molecular patterns (PAMPs) and initiate PAMP-triggered immunity (PTI) [1]. Successful pathogens deliver effectors to the plant apoplast and various intracellular compartments; these effectors suppress PTI and thereby facilitate invasion of the host. As a strategy to counter effectors, plants have evolved intracellular receptors with nucleotide-binding leucine-rich repeat domains that sense effectors and mediate effector-triggered immunity [2].

The bacterial PAMP flg22, a conserved 22-amino-acid peptide derived from *Pseudomonas syringae* flagellin, has provided a powerful tool to decipher PAMP-induced signaling pathways and revealed the complexity of PTI-related mitogen-activated protein kinase (MAPK) cascades. In *Arabidopsis thaliana*, flg22 recognition is mediated by the leucine-rich repeat (LRR) receptor kinase FLAGELLIN SENSITIVE 2 (FLS2). Recognition of flg22 by FLS2 induces an array of defense responses, including the generation of reactive oxygen species (ROS), callose deposition, ethylene production, and reprogramming of host cell genes [1].

Flg22 recognition leads to the activation of two MAPK signaling pathways. One of these MAPK cascades is defined by the mitogen-activated protein kinase kinases (MAPKKs) MKK4 and MKK5, which act redundantly to activate the MAPKs MPK3 and MPK6 [3]. The second flg22-activated cascade is defined by the mitogen-activated protein kinase kinase kinase (MAPKKK), MEKK1. This kinase activates the MAPKKs MKK1 and MKK2, which act redundantly on the MAPK MPK4 [4, 5]. Flg22 also induces the activity of several other MAPKs, but their functions in plant immunity remain to be clarified [6, 7]. Flg22 also transiently activates multiple calcium-dependent protein kinases (CDPKs) in *A. thaliana*. Moreover, four related CDPKs were identified as early transcriptional regulators in PAMP signaling [8]. PAMP-induced protein kinase cascades ultimately lead to the regulation of immune response genes to adjust the metabolic and physiological status of the challenged plants. This regulation can occur at the transcriptional or post-transcriptional level, including AS. Using *mpk3*, *mpk4*, and *mpk6* mutants, we previously confirmed by microarray-based transcriptomics that PTI defense gene expression is strongly regulated by these three MAPKs [9].

AS of mRNAs is important for stress responses in plants [10, 11]. However, very few examples exist how plant signals regulate AS in plant immunity [12–14]. Interestingly, proteomic analyses identified a considerable number of splicing-related proteins as major phosphorylation targets in plants [15], and several of these splicing proteins are phosphorylated by MAPKs *in vitro* [16, 17], suggesting a role of MAPKs in AS regulation. Later work confirmed that

certain splicing factors are phosphorylated in response to PAMP signaling and carry phosphorylation motifs for CDPKs and MAPKs [18]. Recently, we showed that several phosphorylated splicing factors are direct targets of MAPKs [19]. For example, MPK4 targets several phosphorylation sites in SCL30 [19]. Moreover, *mpk4* mutants are compromised in phosphorylation of the splicing factor SKIP and the RNA helicase DHX8/PRP22, both of which are connected to SCL30 in the phosphorylation network of MPK4 [19].

RNA-sequencing (RNA-seq) allows the assessment of differential expression and AS of genes through quantification of transcripts across a broad dynamic range. Transcript expression levels are inferred based on the number of aligned reads and several methods were specifically developed to evaluate AS events at a genome-wide scale [20]. Nevertheless, quantification of transcript isoforms from RNA-seq data remains a substantial challenge. Here, we performed RNA-seq of Arabidopsis treated with flg22. We used standard transcriptome analysis to identify differentially expressed genes (DEGs) and the bioinformatic pipeline AtRTD2 [21] to quantify differential alternative splicing (DAS). These data revealed that 506 genes undergo AS during PTI. However, only 17% of the DAS genes were identified as DEGs. Moreover, DAS genes and DEGs encoded substantially different functional classes of proteins, indicating that current transcriptome analyses miss many regulated transcripts.

Subsequent analysis of *mpk3*, *mpk4*, and *mpk6* mutants for defects in PAMP-triggered AS revealed that *mpk4* mutant plants were strongly compromised in more than 40% of AS upon PAMP treatment, whereas no major changes in AS transcripts were observed in *mpk3* and *mpk6* mutants under these conditions. Several AS targets involved important rearrangements of protein isoforms of the critical PTI regulators CPK28, CRK29, and SERK4. These results show the genome-wide impact of AS in PTI and the key role MAPKs plays in PAMP-induced AS.

## Results

### Flagellin induces alternative splicing in 506 genes

To identify whether PTI signaling was linked to AS processes in plants, we treated Arabidopsis Col-0 plants for 30 min with flg22 or $H_2O$ and analyzed the transcripts by RNA-seq from three biological replicates. An average of 46 million 100-bp long paired-end sequencing reads was obtained for each sample. For quality assessment of the RNA-seq data, we analyzed the proportion of read alignments and the genome-wide sequencing coverage. A comparison of the mapped reads with the annotated genes showed that at least 98% of the reads were from exonic, intronic and 5' or 3' untranslated regions and only 1% mapped to intergenic regions, respectively (S1A Fig). Lastly, we assessed the sequencing saturation and found extensive coverage of the chromosomes (S1B Fig).

We next analyzed the flg22-treated RNA-seq data for DAS events in comparison to mock-treated wild-type plants. We obtained a total of 546 flg22-induced DAS events corresponding to 506 unique DAS genes when considering a p-value cutoff of 0.001 (Fig 1A), with intron retentions as the most frequent events (85%), followed by 7% and 6% alternative 3' and 5' splice sites (3ASS and 5ASS, respectively). Finally, only 6% and 1 single event were obtained for skipped exons (SE) and mutually exclusive exons (MXE), respectively. These results indicate that flg22 induces AS in a considerable number of genes.

When comparing the set of 506 PAMP-induced DAS genes with the set of 1950 DEG (FDR < 0.01, |Fold Change| > 2) (Fig 1B; S1 Data), an overlap of only 89 genes (17%) was obtained. Even when lowering the cutoff value in the set of DEGs to |Fold Change| > 1.5 (S2 Fig), most DAS genes did not overlap with the DEG data set and hence did not appear as PAMP-regulated genes during classical transcriptome analysis. A comparison of the categories

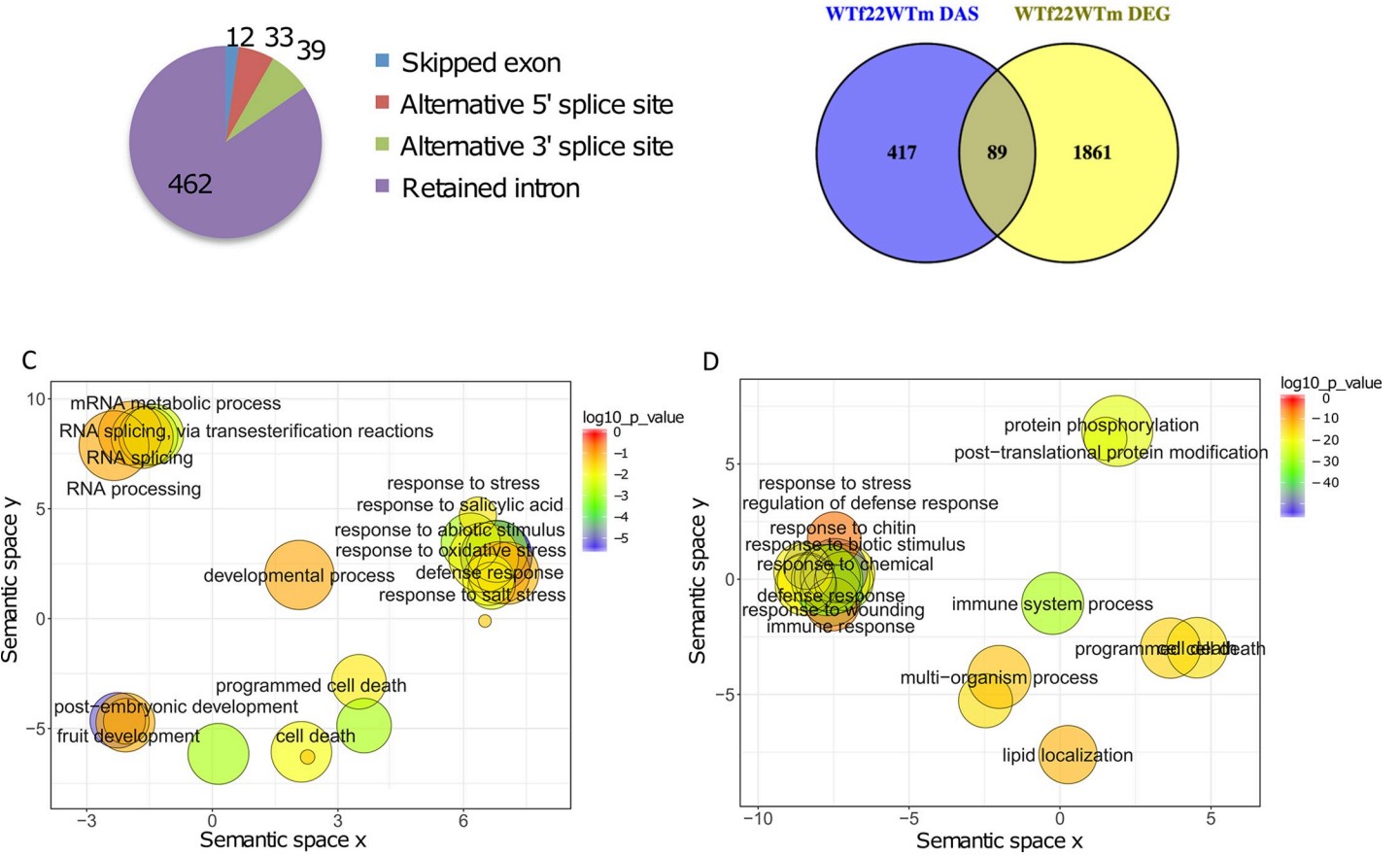

**Fig 1. Flagellin induces alternative splicing in 506 genes. A**, Number of DAS events from each class identified upon flg22 treatment in wild-type plants **B;** Venn comparison plot between differentially spliced (flg22 DAS) with differentially expressed (flg22 DEG) genes in wild-type plants (FDR < 0.01, Fold Change > 2). **C**, REVIGO plots of gene ontology enrichment clusters of differentially spliced, or **D**, differentially expressed genes. Each circle represents a significant GO category but only groups of highest significance are labeled. Related GOs have similar (x, y) coordinates.

of genes that were differentially spliced (DAS) versus those that were differentially expressed (DEG) showed some overlap in the functional gene ontology (GO) categories of metabolic processes (Fig 1C and 1D). Specific enrichment of non-overlapping DAS genes was found in the GO classes of RNA metabolism and development. However, the DAS genes were also enriched for functional categories related to plant defense responses and immunity (S3 Fig).

To reveal the role of RNA metabolism in the set of PAMP-induced DAS genes, we focused on key marker functions in the 506 genes. Indeed, several splicing factors were identified, such as SCL33, SR30, SR45a, RS40, RS41, U2AF65A, RZ1B and RZ1C. Many factors involved in transcriptional regulation, such as RNA-binding proteins, helicases and transcription factors, were found in the set of DAS genes (S2 Data). The bioinformatic identification of various AS events in a selected number of genes involved in immunity or RNA processing was verified by semi-quantitative RT-PCR and PAGE analysis (Fig 2).

## MPK4 is a major regulator of PAMP-induced AS

To assess whether the immune-regulated MAPKs, MPK3, MPK4, or MPK6, play a role in regulating PAMP-triggered AS, RNA-seq data were obtained from the three MAPK knock-out mutants before and after flg22 treatment. When we analyzed *mpk3*, *mpk4*, and *mpk6* in the

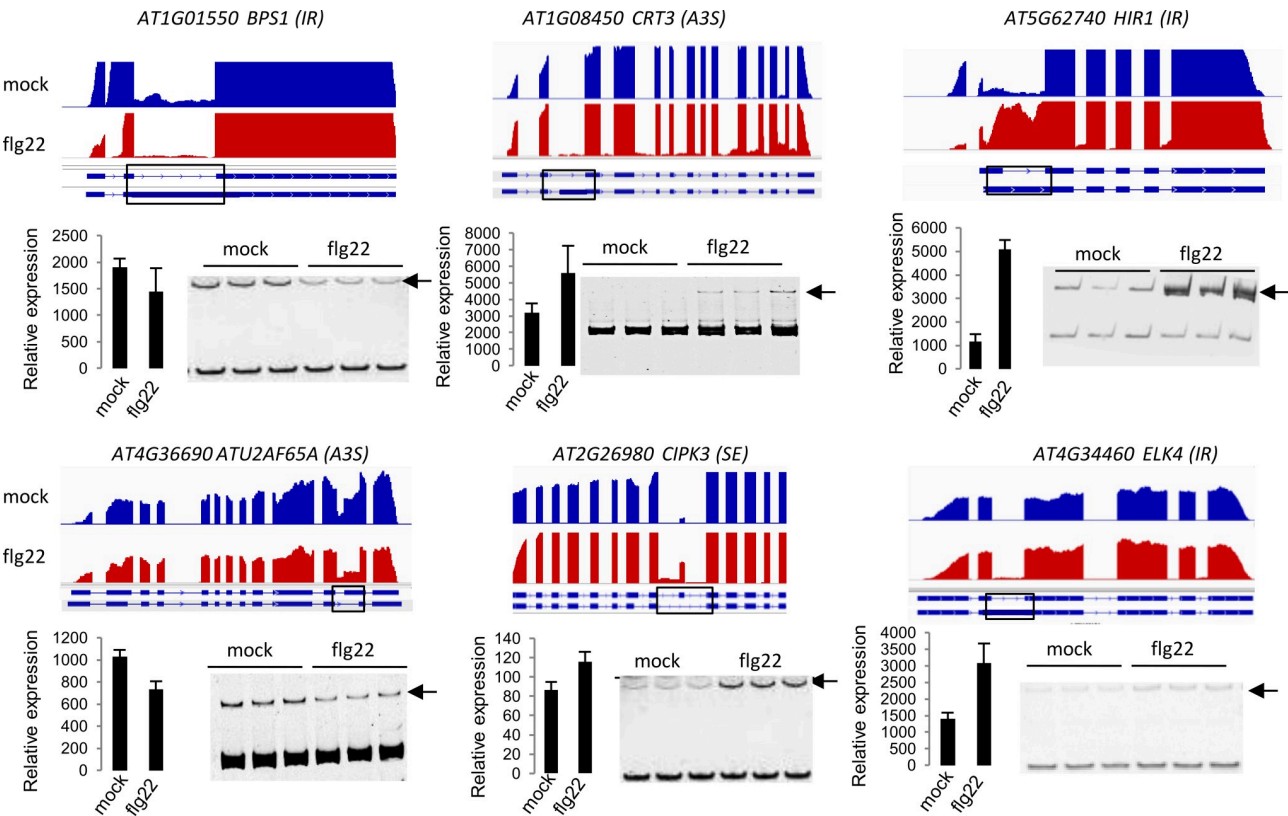

**Fig 2. RT-PCR analysis of selected differential splicing events in response to flg22.** Normalized RNA-seq coverage data are shown in the upper part of each panel. Scale is the same in each track. Relative expression for each gene was calculated based on the normalized RNA-seq read count produced by DEseq2. A black arrow marks the position of the alternative isoform. RT-PCR was performed on 3 biological replicates.

absence of PAMP treatment, no major changes in AS events were observed when compared to wild type plants (Table 1, S4 Fig).

However, upon flg22 treatment, we found that PAMP-induced DAS responses were strongly compromised in *mpk4* but not in *mpk3* or *mpk6* (Fig 3A). Indeed, comparison of the PAMP-induced DAS events observed in *mpk3* and *mpk6* upon flg22 treatment to those in wild-type plants only revealed a total of 11 and 14 DAS genes, respectively (Table 2, S4 Fig). Since MPK3 and MPK6 nonetheless show a considerable number of DEGs, we consider the small number of splicing events as minor and conclude that MPK3 and MPK6 are not involved to a significant extent in regulating PAMP-induced AS (S5 Fig). In contrast, comparing *mpk4* plants to wild-type Col-0 in their response to flg22, we identified 364 differential AS events (Table 2, Fig 3B) that correspond to 344 unique DAS genes (S2 Data). Most of the events belong to the class of intron retention [297] followed by 30 3' alternative splice sites (3ASS), 28

**Table 1. Total number of DEG in mpk4, summ2 and summ2mpk4 as compared to Col-0 in the presence or not of flg22.**

| Number of DEG events (Pval cutoff 0.0001 / FDR 0.05] | |
| --- | --- |
| **Comparison vs Col-0** | **Total DEGs** |
| mpk4 | 3542 |
| summ2 | 93 |
| summ2 mpk4 | 233 |

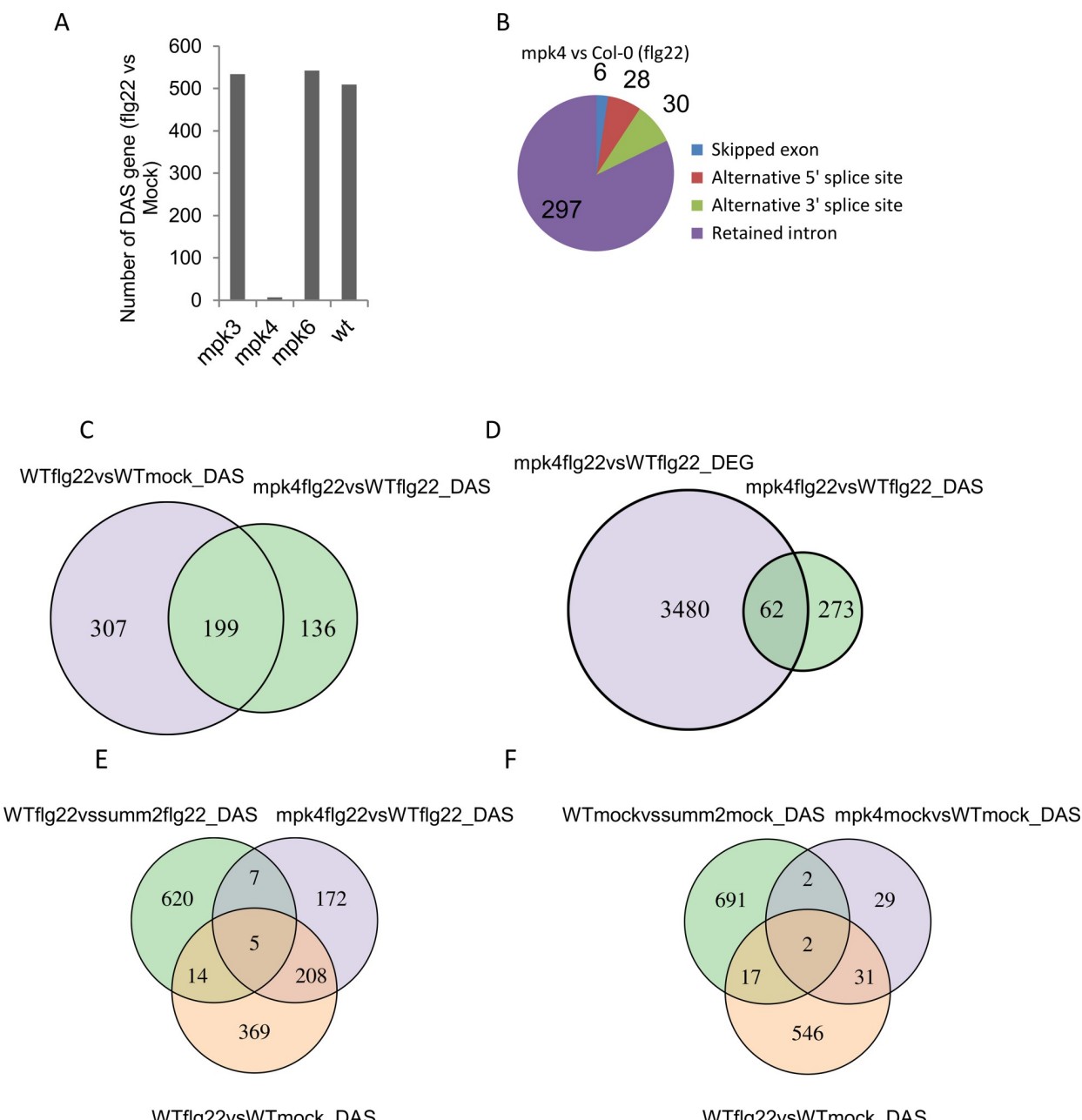

**Fig 3. MPK4 regulates alternative splicing in PTI. A**, Number of DAS genes in response to flg22 in *mpk3*, *mpk4*, *mpk6*, and wild-type (wt, Col-0) plants. **B,** Number of AS events from each class identified in the *mpk4* mutant compared to wild type upon flg22 treatment. **C,** Venn comparison plot between DAS genes in *mpk4* to wild type upon in response to flg22 treatment with DAS genes in mock- and flg22-treated wild-type plants. **D,** Venn comparison plot between DAS genes and DEG genes in *mpk4* as compared to WT. Comparison of flg22-induced DAS events of Col-0 WT with those in *mpk4* and *summ2* upon **E,** mock or **F,** flg22 treatment.

5' alternative splice sites (5ASS), and 6 skipped exons (SE) and 3 mutually exclusive exons (Fig 3B). As analyzed in Venn diagrams, the identity of DAS events in *mpk4* plants strongly over-lapped those seen in flg22-treated wild-type plants (Fig 3C). In addition, a core of 225 PAMP-induced AS events still occurs in *mpk3* and *mpk6* but is lost in *mpk4* (S4A Fig). Out of the total number of 506 flg22-induced DAS genes in wild type, 39% (199 genes) were affected in the

**Table 2. Number of DAS events in *mpk3, mpk4, mpk6, summ2 and summ2mpk4* as compared to Col-0 upon mock (H$_2$O) or flg22 treatment.**

| Number of DAS events (Pval cutoff 0.0001 / FDR 0.05) | | | | | |
|---|---|---|---|---|---|
| Comparisons vs Col-0 | Skipped exon | Mutually exclusive exon | Alternative 5' splice site | Alternative 3' splice site | Retained intron |
| mpk3 (flg22) | 0 | 0 | 0 | 3 | 8 |
| mpk3 (mock) | 0 | 0 | 2 | 0 | 10 |
| mpk4 (flg22) | 6 | 3 | 28 | 30 | 297 |
| mpk4 (mock) | 3 | 0 | 1 | 13 | 45 |
| mpk6 (flg22) | 7 | 0 | 2 | 0 | 5 |
| mpk6 (mock) | 0 | 0 | 0 | 0 | 10 |
| summ2 (flg22) | 41 | 2 | 130 | 117 | 416 |
| summ2 (mock) | 46 | 2 | 113 | 142 | 492 |
| mpk4summ2 (flg22) | 41 | 1 | 131 | 147 | 562 |
| mpk4 summ2 (mock) | 49 | 1 | 71 | 113 | 434 |

*mpk4* knock out mutant upon flg22 treatment (Fig 3C). These results reveal that a major segment of flg22-induced DAS is mediated upon activation of MPK4. Consistent with MPK4's non-immune-related role, 136 DAS genes in *mpk4* mutants were not related to the set of flg22-induced DAS genes in wild-type plants (Fig 3C).

The high rate of intron retentions observed in response to flg22 and in *mpk4* suggested that exogenous PAMP treatment may have a global effect on splicing. Metagene analysis of RNA-seq coverage on introns did not reveal a global trend of intron retentions mediated by flg22 or *mpk4* (S6A Fig). Interestingly, in WT plants, analysis of intron retention levels showed 63% of differentially spliced introns that were more efficiently spliced in response to flg22. On the opposite, 68% and 75% of DAS introns were retained in *mpk4* compared to WT in mock and flg22 treated conditions, respectively (S6B Fig), suggesting that MPK4 is required for splicing of specific introns in response to flg22. In addition, comparison of PAMP-induced and MPK4-dependent IR events revealed that flg22 treatment and the *mpk4* mutation have an opposite effect on a set of 194 common introns [48 retained and 146 spliced introns upon flg22 treatment, S6C Fig).

We next compared the number of DAS genes that overlapped with the DEGs upon treatment of *mpk4* mutants with flg22. As observed in wild-type plants (Fig 1A), only 18% [62] of DAS genes were found in the DEG dataset when using a two-fold change cutoff value (Fig 3D), further confirming that most DAS events are not represented in the set of DEGs in the mutant plants. Consistent with the bioinformatic analyses, we found that, out of the six flg22-dependent DAS genes tested by RT-PCR in Fig 2, five did not show significant flg22-induced AS in the *mpk4* background (Fig 4, S7 Fig). Hence, MPK4 is required for correct flg22-induced AS.

## SUMM2 suppresses DEG but not DAS of MPK4

Recently, loss-of-function mutations in *SUMM2* (suppressor of *mkk1mkk2]* were identified that fully suppress the autoimmune-related growth defects of *mpk4* mutants [22]. *SUMM2* was found to encode a nucleotide-binding leucine-rich repeat (NB-LRR) protein suggesting that the MEKK1-MKK1/2-MPK4 pathway is guarded by the NB-LRR protein SUMM2 to monitor interference by pathogen effectors. Since *summ2* mutants almost completely suppress the strong dwarf and auto-immune phenotype of *mpk4* mutants [22], it might be expected that the massive set of DEG and DAS in *mpk4* might also be suppressed in *summ2* mutants.

We therefore performed RNAseq analysis of *summ2*, *mpk4* and of *mpk4summ2* double mutants. When compared to wild type plants, transcriptome analysis of *summ2* only revealed

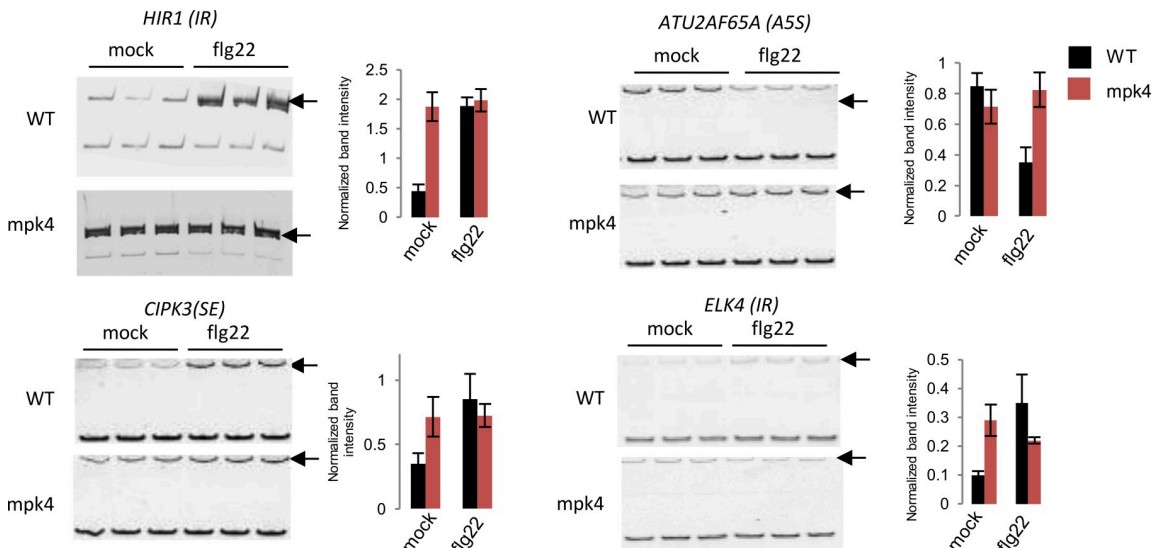

**Fig 4. RT-PCR analysis of selected differential splicing events in *mpk4* and WT in mock- or flg22-treated plants.** RT-PCR was performed on 3 biological replicates and products were separated on the same gel. A black arrow marks the position of the alternative isoforms. Band intensity was normalized against the non-differential isoforms. Significant differences were calculated by t-test (*: p-value < 0.05).

93 genes to be differentially expressed when compared to wild type. This contrasted with the massive set of 3542 DEGs in the *mpk4* mutant (Table 1). As expected, 93% of these differentially expressed genes in *mpk4* were suppressed in the *mpk4summ2* double mutant, showing only 233 DEGs (Table 1). These results are consistent with the genetic analysis that SUMM2 guards MPK4. However, when we analysed the AS events in *summ2*, *mpk4* and *mpk4 summ2*, a different picture emerged. In contrast to the small number of 62 AS events in *mpk4*, *summ2* showed a large number of 795 AS events (Table 2). Moreover, the number of AS events triggered by *summ2* did not change considerably in the *mpk4summ2* double mutant where 673 AS events were detected (Table 2). These results indicate that SUMM2 suppresses the differential gene expression of *mpk4* mutants, but still regulates a large set of AS events independently of MPK4.

We then compared the set of flg22-triggered DAS events in WT with the one induced in *mpk4*, *summ2* or *summ2mpk4* mutants in the presence or absence of flg22 (Fig 3E and 3F). Interestingly, DAS events in *mpk4* and *summ2* were hardly overlapping in the absence or presence of flg22, indicating that MPK4 and SUMM2 regulate AS of different genes. In addition, the set of flg22-triggered DAS events significantly overlapped with the one induced in *mpk4* but not in the *summ2* mutant, suggesting that SUMM2 regulates AS of genes independently of the flg22 response. We also asked how many of the MPK4-dependent events still occur in the *summ2mpk4* mutant in the presence or absence of flg22. We found almost no overlap between *mpk4* and *mpk4summ2* DAS events suggesting that MPK4-dependent AS regulation is independent of SUMM2 (S8 Fig).

It is important to note that most DAS events in *mpk4* plants are triggered only upon flg22 treatment (Table 2), whereas those in *summ2* are already observed in untreated plants. Moreover, the majority of AS events in *mpk4* and *summ2* targets different genes. We conclude from these data that the nucleotide-binding leucine-rich repeat receptor SUMM2, which guards the MPK4 pathway [22], plays a minor role in the regulation of the AS events of flg22-triggered MPK4 activation although controlling MPK4-dependent DEGs.

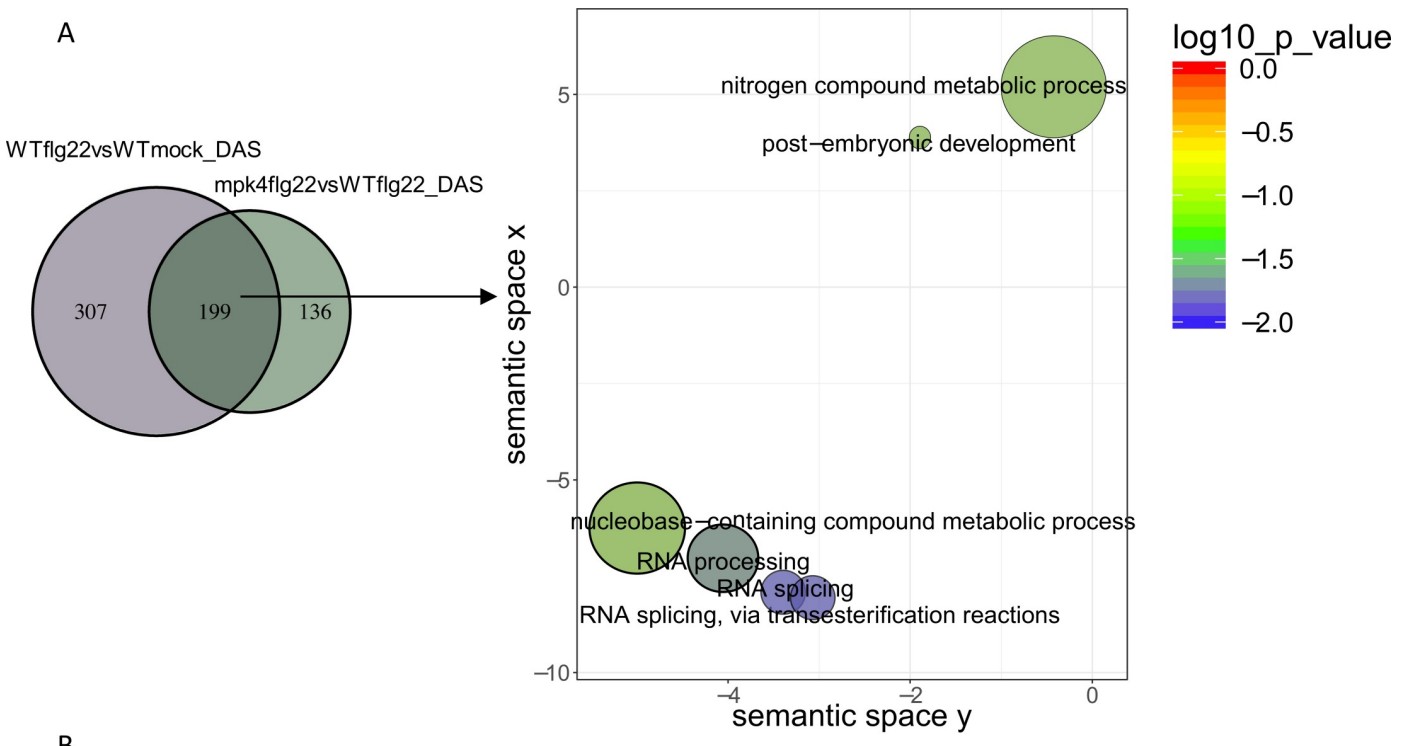

**Fig 5. MPK4 regulates alternative processing of several splicing factors. A,** REVIGO plots of gene ontology enrichment clusters of DAS genes in both *mpk4* compared to wild type and in response to flg22 in wild-type plants. Each circle represents a significant GO category but only groups of highest significance are labeled. Related GOs have similar (x, y) coordinates. **B,** List of DAS genes involved in mRNA processing and splicing.

## MPK4 regulates AS of several splicing factors and key stress signaling genes

GO enrichment analysis of PAMP-induced DAS targets regulated by MPK4 revealed a strong enrichment for RNA processing and splicing categories (Fig 5). A set of factors implicated directly in AS exhibited alternative splice site selection in response to flg22, and this AS was dependent on MPK4. Among the DAS splicing factors, several serine/arginine-rich (SR) splicing factors were identified. SR proteins and SR-related proteins are important regulators of constitutive and alternative splicing and other aspects of mRNA metabolism. In the set of PAMP-induced DAS genes that are dependent on MPK4, we identified four nuclear speckle-localized SR proteins At-SCL33, At-SR31, At-RZ1B and At-SR45a [23–26] (S2 Data).

To analyze the DAS genes affected by MPK4 in relation to pathogen responses, we focused on important immunity-related genes in the list of AS events in *mpk4* (S2 Data). In the list of

PAMP-induced DAS genes that are dependent on MPK4, we identified *RPP4*, which encodes a nucleotide-binding leucine-rich repeat protein with Toll/interleukin-1 receptor domains that confers Arabidopsis resistance to *Peronospora parasitica* [23].

In the MPK4-regulated set of exon-skipping DAS genes, we identified the Ser/Thr protein kinase CIPK3, which associates with a calcineurin B-like calcium sensor, and regulates abscisic acid- and stress-induced gene expression in Arabidopsis [24]. The *cipk3* mutants show altered expression of several markers of abscisic acid, cold and salt stress. Another interesting gene showing exon skipping encodes the transcription factor WRKY26 which plays a role in thermotolerance [25]. Finally, MPK4 also regulates PAMP-induced AS of *NTH2*, encoding a glycosylase-lyase that is involved in oxidative DNA damage repair [26].

In the list of 5ASS genes, *CIPK3* is found again, as well as the MAP kinase gene *MPK17* and the transcription factor gene *WRKY19*, which encodes a MAPKK kinase, but for which no function(s) have yet been attributed.

## Flagelling induces protein kinase transcript isoform switching

Functional consequences of AS rely on changes in the use of different isoforms and on the functional differences between the proteins encoded by alternative isoforms. Therefore, we used a combination of recent methods allowing isoform quantification from RNA-seq read pseudo-counts, statistical identification of changes in isoform abundance ratio (isoform switching events), and automated prediction of putative functional differences between alternative isoforms. When we used this approach on flg22-treated wild-type plants, we identified 68 genes showing significant isoform-switching events (Fig 6A; S3 Data), 40 of which were also identified as DAS by the MATS software. Strikingly, GO analysis of these 68 genes revealed unique enrichment for protein kinases, some of which are involved in PTI (Fig 6B). AS of many of these genes leads to changes in the kinase domains of the encoded proteins. For instance, as shown in Fig 6D, we found isoform switching in the mRNA region encoding the kinase domain of CPK28, a calcium-dependent protein kinase that attenuates PTI by phosphorylating BIK1, which is a cytoplasmic protein kinase-mediating multiple pattern recognition receptors [27]. Similarly, two cysteine-rich receptor-like kinase (CRK) genes, *CRK13* and *CRK29*, (Fig 6E) displayed conserved intron retention events in their kinase domains, leading to increased usage of isoforms containing both serine/threonine and tyrosine kinase domains. Interestingly, these two CRKs differentially influence the sensitivity to infection by pathogenic *Pseudomonas syringae* strains [28]. Finally, *SOMATIC EMBRYOGENESIS-RELATED KINASE4* (*SERK4/BKK1)*, which encodes an FLS2 co-receptor together with its closest homolog BAK1 [29–31], was transcriptionally induced by flg22 but showed a decrease in the isoform predicted to contain a functional kinase domain (Fig 6C). We then used isoform-specific RT-qPCR to confirm isoform switching of *CPK28*, *CRK29* and *SERK4*. The calculation of the relative abundance of the different isoforms confirmed the bioinformatically detected changes (Fig 6F and 6G). To assess whether the flg22-induced AS isoforms could be NMD targets, we analyzed the *upf1upf3* mutant, which is strongly impaired in NMD dependent RNA quality control, for flg22-induced isoform switching of *CPK28*, *CRK29*, and *SERK4* AS using isoform-specific RT-qPCR. The abundance of alternative isoforms of *CRK29* and *SERK4* were unchanged in *upf1upf3* suggesting that these transcripts are not recognized by the NMD machinery. Alternatively, intron-retained isoforms of CPK18 (AT5G66210_P5) accumulated in *upf1upf3* in mock conditions leading to a decrease of its relative abundance versus the main isoform (AT5G66210_P1) but did not seem to be affected upon flg22 treatment. To assess to which extent other PAMP-triggered AS transcripts could be NMD targets, we compared our flg22-induced DAS dataset to that for *upf1upf3* double mutants [24]. As shown in S9 Fig, only

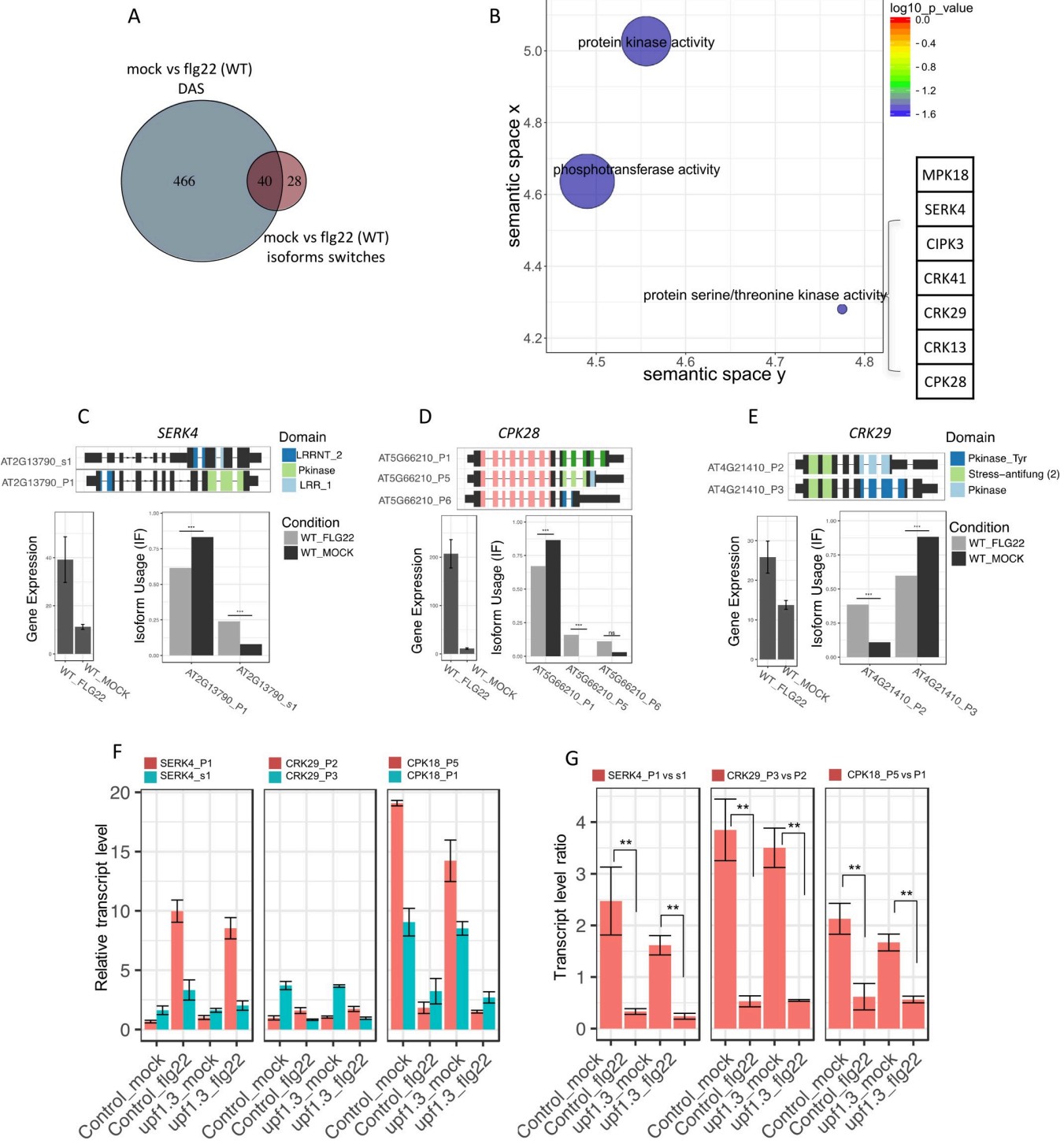

**Fig 6. Flg22-induces protein kinase isoform switching. A**, Venn comparison plot between DAS genes compared to genes with isoform switching events in response to flg22 in WT. **B,** REVIGO plots of gene ontology enrichment clusters of DAS genes both in *mpk4* compared to wild type and in response to flg22 in wild-type plants. Each circle represents a significant GO category but only groups of highest significance are labeled. Related GOs have similar (x, y) coordinates. **C-D-E**, Results of IsoformSwitchAnalysisR showing the gene diagram of differential isoforms with the predicted PFAM domain, change in gene expression and relative isoform abundance in response to flg22 and isoform-specific RT-qPCR analysis showing **F**, the relative transcript abundance (normalized against the housekeeping gene *AT4G26410)* and **G** ratio of isoform abundances in response to mock or flg22 treatment in WT and *upf1upf3* for SERK4, CPK28 and CRK29. Significant differences were estimated by student t-test (** p-value < 0.01).

108 of the 506 flg22-induced DAS transcripts overlapped with the dataset in *upf1upf3* mutants suggesting that a majority of flg22-induced DAS transcripts are not NMD targets.

## Discussion

In higher plants, AS plays a key role in gene expression as shown by the fact that 60–70% of intron-containing genes undergo alternative processing [32, 33]. AS is important in normal growth, development, as well as in abiotic and biotic stress responses in Arabidopsis [10–14], but little is known how plant signals trigger AS events and gene regulation.

### PAMP-triggered AS of defense genes

PAMP-induced protein kinase cascades regulate the expression of immune response genes to adjust the metabolism and physiological status of plants. We compared the set of 1849 PAMP-induced DEG with that of 506 DAS genes. Only 89 of the 506 DAS genes were included in the set of DEGs. Moreover, PAMP-induced DAS and DEG encoded substantially different functional classes of proteins. DAS genes showed unique enrichment for roles in RNA metabolism and transcription, whereas DEGs showed unique functions in responses to various stimuli, including signaling and defense. To reveal the role of RNA metabolism in the PAMP-induced DAS set of genes, we searched for key markers in the 506 DAS genes. Although most transcripts showed intron retention, alternatively spliced 5ASS, 3ASS, and SE transcript isoforms were identified in several splicing factors. These PAMP-induced DAS genes included the splicing factors RS40, RS41, SCL33, SR30, SR45A, U2AF65A, RZ1B, and RZ1C. A similar observation was made by [34] who found that cold-induced DEGs only partly overlapped with DAS genes and have different predicted functions. Whereas DEGs were highly enriched for categories related to cold stress responses, DAS genes were enriched for RNA splicing and RNA binding proteins [34].

Apart from genes with a direct role in RNA metabolism, we identified several PAMP-induced DAS transcripts that encode factors involved in defense and signaling. Leading the list of PAMP-induced DAS genes are receptor-like kinases, such as the cysteine-rich receptor-like kinases *CRK13* and *CRK29*, or the FLS2 co-receptor BKK1/SERK4. RLKs and immune receptors in general seem to be a preferred target of immunity-related AS events in plants [35]. Among these, two DAS isoforms of the TIR-NB LRR receptor N are important for TMV resistance [36] and AS also regulates the barley and rice CC-NB-LRRs Mla13 and Pi-ta, respectively [37–38]. In barley, powdery mildew induced defense is linked to the AS ratio of five different Mla13 splice isoforms [37], whereby in rice, eleven out of the twelve Pi-ta splice variants encode different proteins [38]. In Arabidopsis, RPS4 presents a similar complicated case of six splice isoforms [39]. In many cases, the expression of different isoforms of these receptor-like proteins plays a critical role in defense activation via shaping their intramolecular and intermolecular interactions with themselves and other proteins [35]. However, so far, the underlying molecular mechanisms which control DAS of these genes is poorly understood, but genetics provides some interesting leads. For example, mutation of the SR-type splicing regulator MOS12 affects AS of the Arabidopsis NB-LRRs SNC1 and RPS4 [39]. Besides a direct involvement of splicing regulators, DAS could also be affected by the epigenetic state of the respective genes, their transcription rates and other post-transcriptional events.

Many of the PAMP-induced defense responses are mediated by the activation of transcription factors and several different classes of transcription factors have been found to show PAMP-induced DAS. Among these, the WRKY family plays a key role in abiotic and biotic stresses [40]. Here, we identified *WRKY26* as a target of PAMP-induced AS. WRKY26 functions with WRKY33, which is a key regulator of PTI [1] in response to heat stress [25]. Our

results support the notion that AS of these WRKY transcription factors likely contribute to immunity regulation, a notion that has been proven for the rice WRKYs OsWRKY62 and OsWRKY76 which undergo AS and thereby alter the DNA binding properties and functions in plant defense [41].

Most plant defenses against microbes are based on PAMP-induced production of ROS [42] and a number of enzymes in ROS production contribute directly to disease resistance [43]. Interestingly, besides *CATALASE 3*, a key enzyme in ROS detoxification and a target of virus-induced necrosis [44], we also identified *FQR1* in the set of PAMP-induced DAS genes. FQR1 is a cytosolic quinone reductase that protects Arabidopsis from necrotrophic fungi [45]. Knock-out lines of *fqr1* displayed significantly slower development of lesions of *Botrytis cinerea* and *Sclerotinia sclerotium* in comparison to the wild type. Consistent with a role in disease resistance, *fqr1* mutants displayed increased ROS accumulation and defense gene expression and overexpression of *FQR1* resulted in hypersensitivity to pathogens [45].

## MPK4 is a regulator of alternative splicing in PAMP-triggered immunity

The immune MAPKs MPK3, MPK4, and MPK6, are key players of PAMP-triggered gene expression [1]. We therefore tested whether any of the three mutant MAPKs might also affect these rapid PAMP-induced AS events. In the absence of PAMP treatment, none of the three MAPKs showed significant DAS events when compared to wild type plants, although *mpk4* mutants shows massive derepression of PAMP-regulated DEGs under these conditions. However, in response to flg22, MPK4, but not MPK3 or MPK6, was found to regulate a significant segment of PAMP-induced DAS events, and about 40% of the total number of flg22-triggered DAS events in wild-type plants seem to be regulated by MPK4. Mutants of MPK4 were compromised in AS events of all categories with 85% intron retention, followed by 8% 3ASS and 5ASS, 4% SE and 2% MXE. These results show that the regulation of DEG and DAS events in MPK4 is regulated by different mechanisms. Whereas the absence of MPK4 suffices to derepress PAMP-regulated DEGs, PAMP-induced protein kinase activation of MPK4 is necessary to induce DAS events. The distinct dual functions of MPK4 in the regulation of DEG and DAS events correlate with the absence and presence of PAMP-triggered activation of MPK4, suggesting that MPK4-mediated phosphorylation of specific protein(s) underlies MPK4-induced DAS regulation.

MPK4 is guarded by the NLR SUMM2 [22], and mutations in *summ2* completely suppress the autoimmune phenotype and constitutive expression of defense genes of mutants of *mpk4* [22]. To test whether the DAS set of genes in the *mpk4* mutant is also suppressed by *summ2* mutants, we performed RNAseq of *mpk4*, *summ2* and *mpk4summ2* mutants. In agreement with the suppression of the strong dwarf and cell death phenotype of *mpk4*, the constitutive set of DEGs was almost completely suppressed in the *mpk4summ2* double mutant plants and *summ2* showed only very few DEGs in the absence of PAMP challenge. These data perfectly support the proposed role of SUMM2 to guard the MPK4 pathway [22].

Interestingly, in contrast to MPK4, SUMM2 seems to function as a negative regulator of a large DAS set of genes in the absence of PAMP treatment. However, even upon flg22 activation, the flg22-induced MPK4-dependent DAS gene set was also largely different from that found for the *summ2* mutant. Overall, these data suggest that SUMM2 only guards MPK4 function in terms of DEGs but that the flg22-induced MPK4-dependent DAS events are not regulated by SUMM2. It is, however, possible that these MPK4-dependent PAMP-triggered DAS events are monitored by another NLR system. Among these candidates could be the ANP2/3-MKK6 module, which was also shown to regulate MPK4 immune functions [46]. Since the ANP2/3-MKK6-MPK4 module does not depend on SUMM2, but on the immune

regulators PAD4 and EDS1, certain MPK4 functions might be guarded by distinct NLR systems. The complexity of the MPK4 pathway is also shown by the recent identification of RPS6 as yet another NLR that guards the MEKK1-MKK1/2-MPK4 pathway by specifically surveilling the presence of the HopA1 effector protein of the bacterial pathogen *Pseudomonas syringae* pv. *tomato* [47].

### Phosphorylation of splicing factors in PAMP-triggered AS

A number of splicing-related proteins are PAMP-induced phosphorylation targets of MAPKs *in vitro* and *in vivo* [15, 17–19]. Among these, we identified SCL30 as a target of MPK4. *scl30* mutants are compromised in phosphorylation of the splicing factor SKIP and the RNA helicase DHX8/PRP22, both of which are connected to SCL30 in the phosphorylation network of MPK4 [19]. We also found that flg22-induced AS of the SR protein At-RS31 is dependent on MPK4. AS of *At-RS31* was also shown to be modulated by dark/light transition through a chloroplast retrograde signaling pathway [48], suggesting that it is a central AS event in several signaling pathways. Moreover, *mpk4* mutants were also compromised in flg22-induced splicing of *At-RZ1B*, which interacts with a spectrum of SR proteins [49]. RZ-1B localizes to nuclear speckles and interacts with several SR proteins and loss-of-function of *At-RZ1B* is accompanied by defective splicing of many genes and global perturbation of gene expression [21]. We also found the putative splicing factor At-SR45a, which shows differential 3′ splice site selection in the *mpk4* mutant upon flg22 treatment. SR45a can interact with U1-70K, U2AF [35]b, SR45, At-SCL28, and PRP38-like proteins and undergoes AS itself. U1-70K and U2AF [35]b are splicing factors that play a critical role in the initial definition of 5' and 3' splice sites and in the early stages of spliceosome assembly. SCL28 and PRP38-like protein are homologs of the splicing factors essential for cleavage of the 5' splice site (Tanabe et al., 2009]. The N-terminal extension in the splice form of the SR45A-1a protein inhibits interaction with these splicing factors, suggesting that SR45A helps to form the bridge between 5' and 3' splice sites in the spliceosome assembly and the efficiency of spliceosome formation is affected by the expression ratio of SR45a-1a and SR45a-2 [50].

The findings that MPK4 phosphorylates and at the same time regulates the splicing of several splicing factors adds to the understanding of PAMP-induced DAS. Primary effects of PAMP-triggered signaling by MPK4 on these AS regulators may lead to secondary AS targets, as has been shown for rapid regulation of RS33 by the chloroplast retrograde signaling [48] and for certain auxin targets [51]. This may hint to the existence of yet unknown AS feedback control mechanisms similar to those known for transcriptional circuits.

### Conclusions

Our data suggest that one segment of PAMP-induced responses by MPK4 could be directly regulated by phosphorylation of splicing factors that may generate important isoform changes of key pathogen receptors, signaling components and enzymes controlling plant defense, warranting further investigation into the role of AS regulation in plant defense and signaling.

## Materials and methods

### Plant materials and treatments

*Arabidopsis thaliana* ecotype Col-0 was used as wild type. The MAPK mutants were *mpk4-2* (SALK_056245), *mpk3* (SALK_151594), and *mpk6-2* (SALK_073907). Seeds were surface-sterilised and stratified for 2 d at 4˚C. Seedlings were then grown for 13 d in a culture chamber at 22˚C with a 16 h light photoperiod, on MS plates (0.5 × Murashige Skoog Basal Salts (Sigma

#M6899), 1% sucrose, 0.5% agar, 0.5% MES, pH 5.7). Twenty-four h before treatment, liquid MS (same media without agar) was added to the MS plates to facilitate the transfer of seedlings to liquid MS. Seedlings were treated with deionized water (mock) or with a final concentration of 1 μM flg22 for 30 min and then frozen in liquid nitrogen. In the case of the *mpk4* single mutant, the *mpk4-2* mutation was segregating. These seedlings were thus first grown vertically in MS plates with 1% agar for 7 d to isolate *mpk4-/-* seedlings based on their root phenotype (thickening and shortening of the primary root [82]). Selected seedlings were then grown for another 6 d at 22˚C with a 16 h light photoperiod, on MS plates before transfer to liquid MS and treatments as described above for the other lines.

## RNA extraction, library construction, and sequencing

Three independent biological replicates were produced. For each biological repetition and each timepoint, 14-day-old seedlings grown in long day conditions were collected and RNA samples were obtained by pooling more than 50 plants. Total RNA was extracted with NucleoSpin RNA Plant (MACHEREY-NAGEL), according to the manufacturer's instructions. First-strand cDNA was synthesised from 5 μg of total RNAs using the SuperScript First-Strand Synthesis System for RT-PCR (Life Technologies), according to the manufacturer's instructions. The cDNA stock was diluted to a final concentration of 25 ng/μl. Subsequently, 500 nM of each primer was applied and mixed with LightCycler 480 Sybr Green I Master mix (Roche Applied Science) for quantitative PCR analysis, according to the manufacturer's instructions. Products were amplified and fluorescent signals acquired with a LightCycler 480 detection system. The specificity of amplification products was determined by melting curves. *GADPH* was used as internal control for signal normalization. Exor4 relative quantification software (Roche Applied Science) automatically calculated the relative expression level of the selected genes with algorithms based on the ΔΔCt method. Data were used from duplicates of at least three biological replicates. Sequencing was performed on each library to generate 101-bp paired-end reads on the Illumina HiSeq4000 Genome Analyzer platform. Read quality was checked by the use of FastQC [52] and low quality reads were trimmed by the use of Trimmomatic version 0.32 (http://www.usadellab.org/cms/?page=trimmomatic) with the following parameters: minimum length of 36 bp; mean Phred quality score higher than 30; leading and trailing base removal with base quality below 3; and sliding window of 4:15. After pre-processing the Illumina reads, the transcript structures were reconstructed by the use of a series of programs, namely, TopHat (ver. 2.1.1; http://tophat.cbcb.umd.edu/) for aligning with the genome, and Cufflinks (ver. 2.2.1; http://cufflinks.cbcb.umd.edu/) for gene structure predictions. For TopHat, the Reference-*Arabidopsis thaliana* (TAIR10) genome (https://www.arabidopsis.org) was used as the reference sequences with the maximum number of mismatches set to two. Raw RNA-seq data files are available at the Short Read Archive database (https://www.ncbi.nlm.nih.gov/sra/) under the following accessions numbers: PRJNA379910, PRJNA609667.

## Bioinformatic analysis of RNA-seq data

Reads were quality-checked using FASTQC [52] and trimmed using Trimmomatic V0.36 [53] to remove low-quality reads/bps. Trimmed reads were then aligned to the Arabidopsis reference genome (TAIR10) [54] with AtRTD2 database [21] using Tophat v2.1.1 [55–57]. Differential expression from the aligned reads was calculated using the cufflinks V2.2.1 package [56]. Genes with a two-fold change and a p-value cutoff of 0.05 were considered to be differentially expressed. Genes that were commonly DE and DAS are identified using Venny 2.1.0 [58].

Wild-type, *upf1*, *upf3* and *upf1 upf3* RNA-seq datasets from [59] were downloaded from the ENA (https://www.ebi.ac.uk/ena) under the following accession numbers SRR584115 (WT-1),

SRR584121 (WT-2), SRR584116 (*upf1-1*), SRR584122 (*upf1-2*), SRR584118 (*upf3-1*), SRR584124 (*upf3-2*), SRR584117 (*upf1 upf3-1*) and SRR584123 (*upf1 upf3-2*). Transcript isoform abundance was quantified with pseudo-alignment read count with *kallisto* [60], on all isoforms of the AtRTD2 database [61] and the Araport11 annotation database (https://www.araport.org/data/araport11). Differential expression analysis was performed, both at the transcript or gene level, with DEseq2 with Bonferroni correction of the p-value. Transcripts or genes significantly up-regulated in *upf* mutants compared to Col-0 (p adjusted < 0.01, logFC > 1) were considered as potential NMD targets. The significance of the overlap of NMD targets among DAS genes or isoforms was calculated with a hypergeometric test.

## Alternative splicing analysis

Alternatively spliced genes for wild type and MPK treated/untreated samples were identified using rMATS [62]. Only isoforms with a false discovery rate (FDR) cutoff ≤ 0.05 were considered to be significant. For isoform switching identification, transcript isoform abundance was quantified with pseudo-alignment read count with *kallisto* [60] on all isoforms of the AtRTD2 database [61]. Then the IsoformSwitchAnalyzeR package was used to detect significant changes in isoform usage. Only significant switches (p adj < 0.1) were kept for further analyses [63].

## AS events validation by RT-PCR

Total RNA was treated with DNAseI (Thermo Fisher Scientific) according to the manufacturer's instruction and 1 μg of DNA-free RNA was reversed transcribed with an oligo (dT) primer using the Maxima H Minus Reverse Transcriptase (Thermo). cDNA was amplified with primers spanning the splicing events predicted by rMATS, separated on a 6% PAGE gel, stained for 5 min in SybrGold (Thermo) and imaged using the ChemiDoc XRS system (BioRad). Band pixel density was calculated using ImageJ on three biological replicates. The relative band intensity was calculated as the ratio of the AS RT-PCR product versus the constitutive spliced junction product for each replicate. For isoform-specific RT-qPCR, cDNA was produced as described above and qPCR was performed using primers matching non-overlapping regions between isoforms. The relative abundance of one switching isoform versus an alternative one was calculated using the $2^{ddCt}$ method and normalized against the expression of a housekeeping gene PP2A (AT1G13320) and using the following formula $2^{-((Ct\,ISO1-\,Ct\,PP2A)-(Ct\,ISO2-\,Ct\,PP2A))}$. Primer efficiency was calculated using a dilution series of cDNA. Only primer pairs with efficiency between 95–100% were used for this analysis. Significant differences were calculated using Student's t-test on three biological replicates.

## Supporting information

**S1 Fig. A**, Percentage of mapped reads to exonic, intronic, 5' or 3' untranslated regions (UTR) and intergenic regions in WT upon mock ($H_2O$) or flg22 treatment. **B**, Sequencing coverage of chromosomes.
(TIF)

**S2 Fig. Venn comparison plot between differentially expressed (flg22 DEG) and differentially spliced (flg22 DAS) genes in wild-type plants (FDR < 0.01, Fold Change > 1.5).**
(TIF)

**S3 Fig. Venn comparison plot between differentially spliced (flg22 DAS) with differentially expressed (flg22 DEG) genes in wild-type plants (FDR < 0.01, Fold Change > 2), showing REVIGO plots of gene ontology enrichment clusters of flg22 DAS-specific and flg22 DAS**

**genes that overlap with flg22 DEG.** Each circle represents a significant GO category but only groups of highest significance are labeled. Related GOs have similar (x, y) coordinates.
(TIF)

**S4 Fig.** A, DAS events based on comparison of mock and flg22 in *mpk3*, *mpk4* and *mpk6*. DAS events based on comparison *mpk3*, *mpk4* and *mpk6* as compared to WT upon B, mock and C, flg22 treatment.
(TIF)

**S5 Fig. Venn comparison plot between differentially expressed and differentially spliced genes in *mpk3*, A or *mpk6*, B as compared to Col-0 wild type (FDR < 0.01).**
(TIF)

**S6 Fig. A**, Metagene analysis of all introns in Col-0 and *mpk4* upon mock or flg22 treatment. Read coverage was normalized using counts per million reads (CPM). **B** Number and direction and differential IR events of col-0 WT treated with flg22 or a mock solution. The direction of each splicing events is based on the "inclusion difference" value calculated by rMATS. **C**, Venn diagram showing the overlap of retained and spliced introns in each comparison.
(TIF)

**S7 Fig. RT-PCR analysis of selected differential splicing events in *mpk4* and WT in mock or flg22-treated plants.** RT-PCR was performed on three biological replicates and products were separated on the same gel. Band intensity was normalized against the non-differential isoforms. Significant differences were calculated using a t-test (* p-value < 0.05).
(TIF)

**S8 Fig. Comparison of DAS events in response in *mpk4* and *mpk4summ2* compared to WT upon mock or flg22 treatment.**
(TIF)

**S9 Fig. Differentially regulated isoforms are not enriched for NMD targets.** Comparison of flg22 induced isoforms with the one upregulated in *upf1upf3* in [24].
(TIF)

**S1 Data. Differential gene expression analysis of Col-0, mpk3, mpk4, mpk6 seedlings treated with flg22 or a mock solution.**
(XLSX)

**S2 Data. Differential splicing analysis of Col-0, mpk3, mpk4, mpk6 seedlings treated with flg22 or a mock solution.** DAS genes for each pairwise comparison are shown in a separated tab.
(XLSX)

**S3 Data. Isoforms switching analysis of Col-0, mpk3, mpk4, mpk6 seedlings treated with flg22 or a mock solution.**
(XLSX)

## Author Contributions

**Conceptualization:** Jeremie Bazin, Martin Crespi, Heribert Hirt.

**Data curation:** Jeremie Bazin, Kiruthiga Mariappan, Yunhe Jiang, Thomas Blein.

**Formal analysis:** Jeremie Bazin, Kiruthiga Mariappan, Yunhe Jiang, Thomas Blein.

**Funding acquisition:** Martin Crespi.

**Investigation:** Ronny Voelz, Martin Crespi, Heribert Hirt.

**Methodology:** Jeremie Bazin, Kiruthiga Mariappan, Thomas Blein, Ronny Voelz.

**Project administration:** Martin Crespi, Heribert Hirt.

**Writing – original draft:** Jeremie Bazin, Kiruthiga Mariappan, Martin Crespi, Heribert Hirt.

**Writing – review & editing:** Jeremie Bazin, Martin Crespi, Heribert Hirt.

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
