## [Decision Letter · Decision Letter 0]

14 Jan 2020

Hello Dr. Hirt,

Thank you very much for submitting your manuscript "Role of MPK4 in pathogen-associated molecular pattern-triggered alternative splicing in Arabidopsis" (PPATHOGENS-D-19-02120) for review by PLOS Pathogens. After much effort (that extended across the holiday season), I was able to get reviews of this paper from the same three reviewers that provided comments on the earlier submission. Both reviewer 1 (in confidential comments to me) and reviewer 2 appreciated the importance of adding the demonstration that the DAS phenotype of mpk4 is independent of SUMM2. And both are now supportive of publication. Reviewer 3 remains fairly negative.

The paper lays out an important groundwork and, despite the comments of reviewer 3, I feel that further work at the protein and immune functional level is beyond the scope to be expected here. However, some issues remain. I concur with reviewer 3 that the quantitation does not seem to correspond to the gel images of DAS forms of some genes in figure 4, especially BETA CAS. Also, reviewer 3 lists (and reviewer 1 hints at) inconsistencies between the text and figures and suggestions for improved presentation.

We therefore ask you to modify the manuscript to address the indicated issues raised by reviewer 3 before we can consider your manuscript for acceptance.

(1) A letter containing a detailed list of your responses to the review comments and a description of the changes you have made in the manuscript. Please note while forming your response, if your article is accepted, you may have the opportunity to make the peer review history publicly available. The record will include editor decision letters (with reviews) and your responses to reviewer comments. If eligible, we will contact you to opt in or out.

(2) Two versions of the manuscript: one with either highlights or tracked changes denoting where the text has been changed; the other a clean version (uploaded as the manuscript file).

We hope to receive your revised manuscript within 60 days or less. If you anticipate any delay in its return, we ask that you let us know the expected resubmission date by replying to this email.

[LINK]

Sincerely,

David Mackey

Associate Editor

PLOS Pathogens

Bart Thomma

Section Editor

PLOS Pathogens

Kasturi Haldar

Editor-in-Chief

PLOS Pathogens

orcid.org/0000-0001-5065-158X

Michael Malim

Editor-in-Chief

PLOS Pathogens

orcid.org/0000-0002-7699-2064

Reviewer's Responses to Questions

**Part I - Summary**

Reviewer #1: The paper is very good in the main finding that MAPK4 drives changes in alternative splicing triggered by PAMPs (flg22). This suggests largely separate pathways of regulation of expression of DEGs and DAS genes (transcriptional and AS regulation, respectively. In my opinion, this is expected and reflects cascades of transcriptional and alternative splicing regulation which must be co-ordinated. Identifying MAPK4 as the main driver of PAMP-induced AS is an important step towards dissecting these pathways.

Reviewer #2: A major concern raised by this reviewer was that authors ignores the autoimmunity triggered by the immune receptor in the mpk4 mutant. However, in the revised manuscript by Bazin and co-authors this issue is now addressed and the authors present evidence that MPK4-dependent AS regulation is independent of SUMM2. To address this issue and “spell it out” as the authors now do, is important and strengthens the work significantly. Thus, this reviewer is largely satisfied.

Reviewer #3: Bazin et al. have submitted a revised version of their manuscript entitled “Role of MPK4 in pathogen-associated molecular pattern-triggered alternative splicing in Arabidopsis”. As major changes compared to the initial submission, they have extended the alternative splicing analyses by including the summ2 and the mpk4summ2 mutants. Furthermore, the RNA-seq data are now described in more detail and a more thorough comparison of different samples types is provided. These changes resulted in a better description of the alternative splicing analyses. However, most of the major concerns expressed in the previous round of review have not been addressed, including the mainly descriptive nature of this work. So, I think the authors have some interesting observations, but further research would be needed to address the potential implications of PAMP-triggered alternative splicing changes.

**Part II – Major Issues: Key Experiments Required for Acceptance**

Reviewer #1: In this revised version of the paper, new key experiments have been performed and some data re-analysed and presented and interpreted better.

Reviewer #2: (No Response)

Reviewer #3: 1) The authors describe flg22-induced alternative splicing for genes with potential functions in immunity. However, any evidence for the functional role of these alternative splicing events is missing. Based on their findings, the authors conclude in their summary “These results considerably enlarge the repertoire of protein isoforms that have to be taken into account when studying plant immunity responses.” In Figure 6, protein domains predicted to be encoded in the splicing variants are displayed. Any evidence for the formation of these protein isoforms is lacking. In case of CPK28, the ratio P1 vs P5 changes in the NMD mutant upf3. This is an indicator of NMD targeting, although for a more direct indication, transcript levels and not ratios should be compared. For CRK29, the P3 isoform seems to have a long 3’ UTR (from what can be judged based on the quite small gene models), which is a classical NMD feature. The possible absence of NMD targeting (as mentioned before, the displayed ratios are only an indirect measure) is most likely explained by the absence of translation, as has been shown for many splicing variants before (e.g. Kalyna et al., 2012, NAR). To support their conclusions, the authors would need to show for one or few alternative splicing events that protein isoforms are indeed made (or not).

Even more importantly, the authors would need to examine in at least one case whether the alternative splicing event has any function in immunity.

2) The authors have analysed a set of mutants to address the upstream signalling of flg22-induced alternative splicing. Based on the data presented in the manuscript, the mpk4 mutant seems to play a more important role than mpk3 and mpk6. However, it has to be taken into account that splicing patterns in the mpk4 mutant look already different in the mock sample, at least for some candidates. Figure 4 shows distinct band patterns in comparison of WT and mpk4 under mock conditions for HIR1, BETA CA2, and ELK4. For ATU2AF65A it is difficult to compare as the gel picture for the WT is cut within the lower band. I’m also confused by the quantitation result, which indicates similar ratios for BETA CA2 and ELK4, while the gels show different ratios. Finally, the authors conclude from their analysis of the summ2 mutant that this factor does not play a role in MPK4-induced alternative splicing. Many splicing changes are seen in the summ2 mutant, both in the absence and presence of flg22. Furthermore, almost no overlap between the alternative splicing changes in mpk4 and mpk4summ2 was seen. Given also the large number of alternative splicing changes already observed in the summ2 mutant, I think it is difficult to make conclusions about the specificity of the factors involved in these alternative splicing processes.

**Part III – Minor Issues: Editorial and Data Presentation Modifications**

Reviewer #1: There are still a number of typos and places where the numbers in the text do not match those in Venn diagrams. Please check these carefully.

Reviewer #2: (No Response)

Reviewer #3: 1) The presentation of the RNA-seq data has been improved, but would still require further clarifications and corrections.

For example, are all comparisons in Table 1 relative to WT (mock) or WT (mock/flg22), corresponding to the sample treatment?

Page 4/5: “In contrast, comparing mpk4 plants to wild-type Col-0 in their response to flg22, we identified 419 differential AS events that correspond to 367 unique DAS genes (Table 1B, Fig. 3B).” According to Table 1, sum of AS events is 364.

Page 5, last sentence of first paragraph: according to Fig. 3C there are 136 DAS genes, not 137

2) Fig. 1C, D: text added to circles is difficult to read

3) Fig. 2: in case of BETA CA2, the coverage plots indicate that the longer variant is less abundant than the shorter splicing variant (also under mock condition). The RT-PCR results suggest that the longer variant is more abundant in the mock sample. What is the explanation for these contradictory data? Have all RT-PCR bands been confirmed by sequencing?

4) Fig. 3D: labelling exchanged, DEG set must correspond to larger circle

PLOS authors have the option to publish the peer review history of their article (what does this mean?). If published, this will include your full peer review and any attached files.

Reviewer #1: No

Reviewer #2: No

Reviewer #3: No

---

## [Editor Report · Decision Letter 1]

11 Feb 2020

Dear Dr. Hirt,

We are pleased to inform you that your manuscript 'Role of MPK4 in pathogen-associated molecular pattern-triggered alternative splicing in Arabidopsis' has been provisionally accepted for publication in PLOS Pathogens. And again, my apologies for the delay in response to the last version. The holidays and reviewer unavailability slowed the process significantly.

Before your manuscript can be formally accepted you will need to complete some formatting changes, which you will receive in a follow up email. A member of our team will be in touch within two working days with a set of requests.

Best regards,

David Mackey

Associate Editor

PLOS Pathogens

Bart Thomma

Section Editor

PLOS Pathogens

Kasturi Haldar

Editor-in-Chief

PLOS Pathogens

orcid.org/0000-0001-5065-158X

Michael Malim

Editor-in-Chief

PLOS Pathogens

orcid.org/0000-0002-7699-2064
---

## [Editor Report · Acceptance letter]

27 Mar 2020

Dear Dr. Hirt,

We are delighted to inform you that your manuscript, "Role of MPK4 in pathogen-associated molecular pattern-triggered alternative splicing in Arabidopsis," has been formally accepted for publication in PLOS Pathogens.

Best regards,

Kasturi Haldar

Editor-in-Chief

PLOS Pathogens

orcid.org/0000-0001-5065-158X

Michael Malim

Editor-in-Chief

PLOS Pathogens

orcid.org/0000-0002-7699-2064